# Gender-specific play behavior in relation to autistic traits and behavioral difficulties at the age of seven in the SELMA study

**Fatih Özel** [1,2,3⊛], **Marlene Stratmann** [3⊛] *, **Fotios C. Papadopoulos** [4], **Joëlle Rüegg** [1,3], **Carl-Gustaf Bornehag** [3,5]

1 Department of Organismal Biology, Uppsala University, Uppsala, Sweden, 2 Centre for Women's Mental Health during the Reproductive Lifespan–Womher, Uppsala University, Uppsala, Sweden, 3 Department of Health Sciences, Karlstad University, Karlstad, Sweden, 4 Department of Medical Sciences, Psychiatry, Uppsala University, Uppsala, Sweden, 5 Department of Environmental Medicine and Public Health, Icahn School of Medicine at Mount Sinai, New York, NY, United States of America

⊛ These authors contributed equally to this work.
* Marlene.Stratmann@kau.se

**Citation:** Özel F, Stratmann M, Papadopoulos FC, Rüegg J, Bornehag C-G (2024) Gender-specific play behavior in relation to autistic traits and behavioral difficulties at the age of seven in the SELMA study. PLoS ONE 19(8): e0308605. https://doi.org/10.1371/journal.pone.0308605

**Data Availability Statement:** These data cannot be made publicly available, as the data contains sensitive personal information as well as potentially identifying information. Further, the participants in the study only gave informed consent for

## Abstract

### Background

Childhood gender nonconformity is related to psychological distress and behavioral difficulties. Similarly, there is evidence for a link between gender nonconformity, or gender dysphoria in some studies, and autism spectrum disorder and related traits. Our knowledge on those associations mostly originates from clinical populations, which might lead to overestimation. Thus, this study aimed to assess associations between gender nonconformity and behavioral difficulties in a population-based study.

### Methods

In the Swedish Environmental Longitudinal, Mother and Child, Asthma and Allergy (SELMA) study, cross-sectional associations between gender-specific play behavior and behavioral outcomes and autistic traits were investigated among 718 children at 7-years of age. Play behavior was measured using the Preschool Activities Inventory; behavioral outcomes and autistic traits were measured with the Strengths and Difficulties Questionnaire and the Social Responsiveness Scale, respectively. Linear and logistic regression analyses were performed.

### Results

Higher composite play behavior scores (indicating either increased masculine or decreased feminine play behavior) were associated with increased autistic trait scores in girls ($\beta = 0.13$; 95% confidence interval [CI] = 0.00, 0.26). Furthermore, higher composite scores were shown to be associated with behavioral difficulties in both girls ($\beta = 0.11$; 95% CI = 0.04, 0.18) and boys ($\beta = 0.10$; 95% CI = 0.02, 0.19). Additionally, higher feminine scores were related with increased problems in peer relationships in boys ($\beta = 0.04$; 95% CI = 0.00, 0.07).

researchers employed or affiliated to Karlstad University to use the data. According to the General Data Protection Regulation, the Swedish law, the Swedish Data Protection Act, the Swedish Ethical Review Act, and the Public Access to Information and Secrecy Act, these types of sensitive data can only be made available for specific purposes, including research, that meets the criteria for access to this type of sensitive and confidential data as determined by a legal review. In case of questions please contact Dr. Huan Shu (Huan. Shu@kau.se) who is the administrator for the SELMA data as well as the research ethics committee at Karlstad University (lotta. utterberg@kau.se).

**Funding:** This work received funding from the European Union's Horizon 2020 research and innovation program under grant agreement no. 634880, EDC-MixRisk and the funders had no role in study design, data collection and analysis, decision to publish, or preparation of the manuscript.

**Competing interests:** The authors have declared that no competing interests exist.

## Conclusions

This study suggests a link between gender nonconforming play behavior and autistic traits as well as behavioral difficulties among children in a non-clinical population, which calls attention to the necessity of supporting children with gender nonconformity from early ages.

## Introduction

Gender nonconformity (GNC) refers to variations in gender expression from societal and cultural gender norms [1]. Childhood GNC can manifest itself in several ways; for instance, in play behavior and peer relationships, dressing, body language, and so forth. GNC is different from gender dysphoria (GD), which is defined as a marked incongruence between one's experienced/expressed gender and assigned sex at birth according to the Diagnostic and Statistical Manual of Mental Disorders, Fifth Edition, Text Revision (DSM-5-TR) [2]. Even though expressing childhood GNC does not necessarily indicate developing GD later in life [3], those two phenomena are evidently linked, and the majority of the existing diagnostic criteria of childhood GD are constructed based on GNC [2]. Furthermore, higher levels of recalled GNC have been observed among individuals with GD [4–6]. It is challenging to obtain information on the proportion of individuals with GNC and/or GD due to dissimilarities in definitions and methodologies used in the field [7]. Still, the proportion of transgender and gender diverse persons, i.e., people whose gender identities and/or expressions are different from sociocultural norms, in the general population is estimated to be between 0.02 and 8.4% [7].

In GD research, the high co-occurrence with autism spectrum disorder (ASD) has increasingly drawn researchers' attention in the last decade. In 2010, de Vries et al. first reported a higher prevalence of ASD among children and adolescents with GD in comparison to the general population [8]. In the following years, many studies revealed a higher prevalence of autistic traits or ASD among children and adolescents with GNC, GD or gender diversity [9–15]. A recent cohort study including 11,251 children showed that gender specific play was not different between children without and children with ASD in early childhood [16]. However, in repeated measurements of gender-specific play later in childhood (30 months vs 57 months), boys with ASD showed significantly less gender-specific play behavior than their non-autistic peers, while girls with ASD showed higher gender-specific play compared to the girls without ASD although the result for girls was not significant [16]. Inversely, researchers also focused on individuals with ASD and showed increased levels of GNC, GD and gender diversity among them [17–20]. Even though the evidence for GNC is considerably less than for GD and the majority of studies addressing this co-occurrence has been conducted within clinical populations, there are studies revealing associations between autistic traits and GNC in the general population [10, 21]. For instance, Nabbijohn et al. reported an association between gender variance and ASD-related behaviors among non-clinical children aged 6–12 years [21]. Recalled childhood GNC has also recently been linked to ASD and autistic traits in adults [22, 23]. Recent meta-analyses and systematic reviews display the co-occurrence clearly [24, 25] and the potential explanations for the link between ASD and GD have been reviewed elsewhere [26]. Nevertheless, there have also been concerns raised about methodology used in existing studies showing co-occurrence of ASD and GD and related features [27, 28].

In parallel, a higher risk of experiencing behavioral difficulties has been repeatedly reported among children with GNC [3]. However, evidence on the increased risk is mostly gathered from clinically referred children, and the risk of having behavioral and emotional difficulties

concerning GNC might be different in population-based settings, as the awareness of behavioral difficulties might be different in population samples compared to clinical samples. In a few studies aiming to address this issue, GNC and gender diversity have been associated with internalizing and externalizing problems [29, 30], non-suicidal self-injurious behavior and suicidal thoughts [31] and other behavioral outcomes including social and attention problems and aggressive behavior [32]. Moreover, non-binary youth, i.e., youth that identifies outside the gender binary, have also been shown to have more behavioral difficulties in comparison to binary youth [33].

Overall, there is evidence on the associations between GNC and autistic traits, ASD and behavioral difficulties; however, population-based studies conducted with young children that shed light on the associations are scarce, and clinical study samples might lead to an overestimation of the associations. Besides, both GNC and ASD as well as related behavioral outcomes are fairly heterogeneous and multidimensional, which makes drawing general conclusions even more difficult. In this study, we focused on a single domain of GNC, namely gender-specific play behavior, and aimed to assess its relation to autistic traits and behavioral difficulties at seven years of age using a population-based cohort.

## Methods

### Study design and population

A cross-sectional study design was used to examine the relation between gender-specific play behavior and autistic traits and behavioral difficulties in 7-year-old children.

The Swedish Environmental Longitudinal, Mother and Child, Asthma and Allergy (SELMA) study is a longitudinal and ongoing pregnancy cohort in Värmland county, Sweden. All women in the study were invited during their first antenatal care clinic visit (median 10th week of gestation) to participate in an initial data collection [34]. Eligibility criteria were being able to understand written Swedish and not moving or living outside the county during the study period [34]. Initially, 6,658 women were invited to participate of which 2,582 (39%) agreed to participate [34]. The recruitment period lasted from 1st of November 2007 to 31st of March 2010, and when the child was approximately 7.5 years old, a follow-up health examination was conducted for about 1,000 children. A questionnaire was sent out to these families in order to collect data on their children's play behavior and behavioral outcomes. A more detailed description of the data collection and recruitment has been described earlier by Bornehag et al. [34].

The study population for the current analyses includes 718 children for which full information on all variables is available. The parents provided written informed consent for themselves and their child, and the study was ethically approved by the Regional Ethic Review Board in Uppsala, Sweden (No. 2007/062 and No. 2015/177).

### Gender-specific play behavior

The Preschool Activities Inventory (PSAI), a validated parent-rated questionnaire, was used to determine gender-specific play behavior [35, 36]. The PSAI consists of 12 "feminine" and 12 "masculine" items that are grouped into three categories: toys, activities and child characteristics. Each item is responded to on a five-point scale with the answer options being "1—never", "2—hardly ever", "3—sometimes", "4—often" and "5—very often". Feminine and masculine scores are calculated by summing up all feminine and masculine item scores, respectively. Higher feminine or masculine scores reflect more gender-specific behavior. Furthermore, a composite score is calculated using the following formula: *48.25 + 1.1 x (masculine score—feminine score)*, which was established in an early standardization study [35]. Thus, the composite

score takes both feminine and masculine items into consideration, and higher scores can be interpreted in the direction of increased masculine or decreased feminine play behavior.

Regarding the children's gender-specific play behavior, the parents were requested to fill out a slightly modified 23-item version of PSAI (see the Questionnaire in S1 Questionnaire) in the course of the follow-up wave at the mean age of 7.5 years. Each item from the questionnaire was examined to determine if it was able to distinguish girls and boys in the study population. There was no statistical difference in the mean scores of the feminine item "avoid taking risks" and the masculine item "climbing" between girls and boys. Therefore, feminine, masculine, and composite scores were calculated without those two items.

## Behavioral outcomes

The Social Responsiveness Scale (SRS), version 1, was used to assess social behaviors such as social awareness, social cognition, social communication, social motivation and restrictive interests and repetitive behaviors in the child. Each of the 65 items can be rated on a scale from "0—not true" to "3—almost always true", and is intended to be filled out by parents or teachers [37, 38]. Total scores range from 0 (highly socially competent) to 195 (severely socially impaired). To control the SRS scores for sex, T-scores have been proposed. These are standardized scores based on the raw SRS scores for each child, making the comparison between girls and boys valid. T-scores above 60 indicate deficiencies in reciprocal behavior that might be of clinical relevance [38]. The SRS has shown to be a valid instrument to detect autistic traits in clinically ascertained as well as population-based samples [37, 38]. Moreover, it has been shown to have excellent agreement with the gold standard "Autism Diagnostic Interview—Revised", and it is stable over time and unrelated to age and IQ [38].

The Strengths and Difficulty Questionnaire (SDQ) is an instrument that can be used to screen for behavioral difficulties in children [39]. The questionnaire contains 25 questions, five questions each for emotional symptoms, conduct problems, hyperactivity and inattention, peer relationship problems and prosocial behavior [39]. Answers are scored on a three item scale, which are "0—not true", "1—somewhat true" and "2—certainly true" [39]. Each difficulty subscore (emotional symptoms, conduct problems, hyperactivity and inattention and peer relationship problems) can range from 0 to 10 points, where a higher score indicates more difficulties, except for the prosocial behavior subscore that is reversely scored, where a higher score indicates higher prosocial behavior [39]. Further, a total difficulty score can be calculated using the four difficulty subscores and adding them up to a score between 0 and 40 points [39]. Using a 90th percentile approximation cut-off ($\geq$12 points), associations with potentially clinical relevance can be estimated. Goodman et al. proposed a cut-off of 14 points using the 90th percentile to identify clinically relevant cases in another study population [39]. As the distribution and range of SDQ scores might differ in different populations, the 90th percentile cut-off was chosen in this study instead of the previously proposed scores.

## Covariates

Covariates have been chosen *a priori* based on previously published literature and hypothesized confounders are shown in a directed acyclic graph (DAG) in S1 Fig. Based on this, four variables were included as potential confounders, and all regression models were stratified by sex. The four confounders are child age, maternal age, maternal education and parental attitudes toward children's play behavior. All covariates were collected using self-administered questionnaires. Maternal education was categorized into elementary school, high school, university or other education that is not corresponding to the three previous categories. Parental attitudes toward their children's gender-specific play behavior was assessed with a scale

consisting of 23 "feminine", "masculine" and "gender neutral" items. Parents were asked to respond to the questions "How likely is it that you would buy the following items for your child?" or "How likely is it that you would encourage your child to the following activities?" for each item on a five-point scale from "0—absolutely not" to "4—highly likely" (see the Questionnaire in S2 Questionnaire). Seventeen feminine and masculine items were selected and summed up separately to construct two variables which either indicate "encouraging the child toward feminine play behavior" or "encouraging the child toward masculine play behavior".

## Statistical analysis

Gender-specific play behavior was used as three different continuous scores that are feminine, masculine and composite scores. Separate models were performed for each play behavior score, and all associations of interest were examined in crude and adjusted models in which child age, maternal age, maternal education level and parental attitudes toward play behavior were adjusted for. All analyses were stratified by sex.

The associations between gender-specific play behavior and autistic traits and behavioral difficulties were examined using the SRS total T-score, SDQ total score and their subscales, respectively. Linear regression analysis was performed. Following, using logistic regression analysis, the associations between gender-specific play behavior and potential clinically relevant autistic traits and behavioral difficulties were investigated using the 90th percentile cut-off score ($\geq$12 points) for SDQ and the cut-off score of 60 for SRS. 95% confidence intervals were presented for all analyses. Collinearity plots are shown in S2–S5 Figs.

All analyses were performed using R, version 4.2.2.

## Results

Table 1 shows the population characteristics of the 718 children included in the study, 354 of which were girls (49.3%). The mean ages for both girls and boys were 7.5 years with a standard deviation (SD) of 0.3. Mothers of girls were on average 30.9 (SD 4.9), mothers of boys on average 31.0 years old at baseline (SD 4.5). Considering the whole study population, most mothers had a university degree (around 65.2%). The PSAI feminine subscore was higher in girls (32.2, SD 6.0) than in boys (17.7, SD 4.3), whereas the masculine and composite scores were higher in boys (31.8, SD 6.1 and 63.8, SD 6.7) than in girls (23.1, SD 5.4 and 38.2, SD 8.0). Parents were encouraging their sons toward masculine play behavior more than feminine play behavior and vice versa for their daughters. The SRS total T-score was higher in girls (46.3, SD 8.1) than in boys (45.3, SD 8.5), whereas the total score of the SDQ were higher in boys (6.0, SD 4.7) than in girls (5.4, SD 4.4). More information on the comparisons of PSAI, SRS, and SDQ scores between girls and boys are presented in S1 Table.

Table 2 shows the adjusted associations between gender-specific play behavior and autistic traits. Higher composite scores were significantly associated with increased SRS total T-score ($\beta$ = 0.13; 95% CI = 0.00, 0.26), social awareness ($\beta$ = 0.17; 95% CI = 0.00, 0.34) and social cognition ($\beta$ = 0.14; 95% CI = 0.01, 0.28) subscores in girls. No significant association was observed between play behavior scores and SRS and its subscales in boys.

The results of the crude models for the associations between gender-specific play behavior scores, autistic traits and behavioral difficulties are presented in S2 and S3 Tables.

Adjusted associations between gender-specific play behavior scores and behavioral difficulties are summarized in Table 3. The results show that having higher composite scores in girls was significantly associated with a higher SDQ total score ($\beta$ = 0.11; 95% CI = 0.04, 0.18), more conduct problems ($\beta$ = 0.04; 95% CI = 0.02, 0.06) and more hyperactivity and inattention ($\beta$ = 0.07; 95% CI = 0.04, 0.011), whereas it was associated with decreased prosocial behavior ($\beta$ =

**Table 1. Descriptive characteristics of the study population (N = 718).**

|  | Girls (N = 354) | Boys (N = 364) | p-value | Cohen's d |
|---|---|---|---|---|
| Child age, mean in years, (SD) | 7.5 (0.3) | 7.5 (0.3) | 0.97 | 0.003 |
| Mother age at birth, mean in years, (SD) | 30.9 (4.9) | 31.0 (4.5) | 0.78 | 0.02 |
| Maternal education, N, (%) |  |  | 0.21 | NA |
| Elementary school | 3 (0.8%) | 11 (3.0%) |  |  |
| High school | 99 (28.0%) | 98 (26.9%) |  |  |
| University | 233 (65.8%) | 235 (64.6%) |  |  |
| Other education | 19 (5.4%) | 20 (5.5%) |  |  |
| PSAI feminine subscore, mean, (SD) | 32.2 (6.0) | 17.7 (4.3) | <0.001 | 2.78 |
| PSAI masculine subscore, mean, (SD) | 23.1 (5.4) | 31.8 (6.1) | <0.001 | 1.53 |
| PSAI composite score, mean, (SD) | 38.2 (8.0) | 63.8 (6.7) | <0.001 | 3.49 |
| Parental attitudes feminine subscore, mean, (SD) | 23.5 (6.1) | 18.4 (6.9) | <0.001 | 0.78 |
| Parental attitudes masculine subscore, mean, (SD) | 18.0 (6.0) | 24.3 (5.6) | <0.001 | 1.09 |
| SRS, total T-score, mean, (SD) | 46.3 (8.1) | 45.3 (8.5) | 0.10 | 0.12 |
| SRS, social awareness, mean, (SD) | 49.3 (10.5) | 48.2 (9.7) | 0.15 | 0.11 |
| SRS, social cognition, mean, (SD) | 45.7 (8.4) | 45.0 (8.9) | 0.31 | 0.08 |
| SRS, social communication, mean, (SD) | 46.0 (7.5) | 45.0 (7.9) | 0.07 | 0.14 |
| SRS, social motivation, mean, (SD) | 48.6 (7.4) | 47.6 (8.3) | 0.09 | 0.13 |
| SRS, RIRB, mean, (SD) | 46.2 (8.9) | 45.4 (8.5) | 0.22 | 0.09 |
| SDQ, total score, mean, (SD) | 5.4 (4.4) | 6.0 (4.7) | 0.08 | 0.13 |
| SDQ, emotional symptoms, mean, (SD) | 1.6 (1.8) | 1.4 (1.6) | 0.08 | 0.13 |
| SDQ, conduct problems, mean, (SD) | 1.1 (1.4) | 1.1 (1.3) | 0.56 | 0.04 |
| SDQ, hyperactivity/inattention, mean, (SD) | 2.1 (2.2) | 2.8 (2.4) | <0.001 | 0.30 |
| SDQ, peer relationships problems, mean, (SD) | 0.6 (1.1) | 0.8 (1.3) | 0.04 | 0.15 |
| SDQ, prosocial behavior, mean, (SD) | 8.9 (1.5) | 8.6 (1.6) | 0.005 | 0.21 |

SD, standard deviation; SRS, Social Responsiveness Scale; RIRB, Restricted interests and repetitive behavior; SDQ, Strengths and Difficulties Questionnaire

-0.04; 95% CI = -0.06, -0.01). Additionally, in girls, having higher feminine scores was significantly associated with decreased conduct problems (β = -0.03; 95% CI = -0.06, -0.01) and increased prosocial behavior (β = 0.04; 95% CI = 0.01, 0.07) whereas having higher masculine scores was significantly related to increased hyperactivity and inattention in the SDQ (β = 0.06; 95% CI = 0.01, 0.11). In boys, positive associations of composite play behavior score with the total SDQ score (β = 0.10; 95% CI = 0.01, 0.18), conduct problems (β = 0.02; 95%

**Table 2. Adjusted associations between gender-specific play behavior scores and autistic traits (N = 718).**

|  | SRS total T-score | Social awareness | Social cognition | Social communication | Social motivation | RIRB |
|---|---|---|---|---|---|---|
|  | beta (95% CI) | | | | | |
| Girls | | | | | | |
| Feminine score | -0.14 (-0.29, 0.01) | -0.16 (-0.36, 0.04) | -0.13 (-0.28, 0.03) | -0.11 (-0.25, 0.02) | -0.11 (-0.25, 0.04) | -0.10 (-0.27, 0.07) |
| Masculine score | 0.02 (-0.15, 0.19) | 0.10 (-0.12, 0.32) | 0.03 (-0.15, 0.21) | -0.01 (-0.16, 0.15) | -0.03 (-0.19, 0.13) | 0.05 (-0.14, 0.24) |
| Composite score | **0.13 (0.00, 0.26)** | **0.17 (0.00, 0.34)** | **0.14 (0.01, 0.28)** | 0.10 (-0.02, 0.22) | 0.07 (-0.06, 0.19) | 0.11 (-0.03, 0.26) |
| Boys | | | | | | |
| Feminine score | 0.04 (-0.18, 0.26) | -0.03 (-0.28, 0.22) | 0.01 (-0.23, 0.23) | -0.03 (-0.23, 0.17) | 0.18 (-0.03, 0.40) | 0.08 (-0.14, 0.30) |
| Masculine score | 0.09 (-0.06, 0.25) | 0.10 (-0.07, 0.28) | 0.06 (-0.11, 0.22) | 0.05 (-0.10, 0.19) | 0.07 (-0.08, 0.22) | 0.13 (-0.03, 0.28) |
| Composite score | 0.07 (-0.08, 0.23) | 0.14 (-0.03, 0.32) | 0.04 (-0.12, 0.20) | 0.06 (-0.08, 0.20) | -0.01 (-0.17, 0.14) | 0.08 (-0.08, 0.24) |

**Table 3. Adjusted associations between gender-specific play behavior scores and behavioral difficulties (N = 718).**

|  | SDQ total score | Emotional symptoms | Conduct problems | Hyperactivity/ inattention | Peer relationship problems | Prosocial behavior |
|---|---|---|---|---|---|---|
|  |  |  | beta (95% CI) |  |  |  |
| Girls |  |  |  |  |  |  |
| Feminine score | -0.08 (-0.17, 0.00) | -0.00 (-0.04, 0.03) | **-0.03 (-0.06, -0.01)** | -0.03 (-0.08, 0.01) | -0.01 (-0.03, 0.01) | **0.04 (0.01, 0.07)** |
| Masculine score | 0.08 (-0.02, 0.17) | -0.02 (-0.06, 0.02) | 0.03 (-0.00, 0.06) | **0.06 (0.01, 0.11)** | 0.01 (-0.01, 0.04) | -0.01 (-0.04, 0.02) |
| Composite score | **0.11 (0.04, 0.18)** | -0.01 (-0.04, 0.02) | **0.04 (0.02, 0.06)** | **0.07 (0.04, 0.11)** | 0.01 (-0.01, 0.03) | **-0.04 (-0.06, -0.01)** |
| Boys |  |  |  |  |  |  |
| Feminine score | 0.03 (-0.09, 0.15) | 0.01 (-0.03, 0.05) | -0.02 (-0.05, 0.01) | 0.00 (-0.06, 0.06) | **0.04 (0.00, 0.07)** | 0.04 (-0.00, 0.08) |
| Masculine score | **0.10 (0.02, 0.19)** | 0.02 (-0.01, 0.05) | 0.01 (-0.01, 0.04) | **0.05 (0.01, 0.10)** | 0.02 (-0.00, 0.04) | 0.01 (-0.02, 0.04) |
| Composite score | **0.10 (0.01, 0.18)** | 0.02 (-0.01, 0.05) | **0.02 (0.00, 0.05)** | **0.05 (0.01, 0.10)** | 0.00 (-0.02, 0.02) | -0.01 (-0.04, 0.02) |

CI = 0.00, 0.05) and hyperactivity and inattention (β = 0.05; 95% CI = 0.01, 0.10) were shown. Higher feminine scores in boys were significantly linked to peer relationships problems (β = 0.04; 95% CI = 0.00, 0.07), whereas higher masculine scores were significantly related to the SDQ total score (β = 0.10; 95% CI = 0.02, 0.19) and hyperactivity and inattention (β = 0.05; 95% CI = 0.01, 0.10).

In Table 4, the results of the adjusted associations of gender-specific play behavior with clinically relevant autistic traits and behavioral difficulties are shown. Higher feminine scores were associated with decreased likelihood of having autistic traits (Odds Ratio [OR] = 0.90, 95% CI = 0.82, 0.98) and less behavioral difficulties (OR = 0.93, 95% CI = 0.87, 0.99) in girls. Moreover, higher composite scores significantly increased the likelihood of having behavioral difficulties (OR = 1.07, 95% CI = 1.02, 1.13) among girls. The results of the unadjusted associations are presented in S4 Table.

## Discussion

In this population-based study, we examined the associations between gender-specific play behavior and autistic traits and behavioral difficulties in 7-year-old children. Regarding autistic traits, we showed an association between composite play behavior score and SRS total T-score as well as social awareness and social cognition subscores in girls. Furthermore, results concerning behavioral difficulties demonstrated that higher composite scores were associated with increased behavioral difficulties in general, particularly for conduct problems and hyperactivity and inattention in both girls and boys. Higher masculine play behavior scores were similarly associated with hyperactivity and inattention in both sexes. In boys, higher feminine

**Table 4. Adjusted associations between gender-specific play behavior scores and clinically relevant behavioral outcomes and autistic traits (N = 718).**

|  | SRS lower cut-off | SRS higher cut-off | SDQ 90th percentile |
|---|---|---|---|
|  |  | OR (95% CI) |  |
| Girls |  |  |  |
| Feminine score | **0.90 (0.82, 0.98)** | 0.86 (0.69, 1.10) | **0.93 (0.87, 0.99)** |
| Masculine score | 0.95 (0.87, 1.05) | 1.03 (0.82, 1.29) | 1.02 (0.95, 1.09) |
| Composite score | 1.03 (0.97, 1.11) | 1.30 (0.91, 1.86) | **1.07 (1.02, 1.13)** |
| Boys |  |  |  |
| Feminine score | 1.05 (0.92, 1.18) | 1.22 (0.99, 1.51) | 1.07 (0.98, 1.17) |
| Masculine score | 1.02 (0.94, 1.12) | 1.11 (0.95, 1.28) | 1.06 (0.99, 1.13) |
| Composite score | 1.00 (0.92, 1.09) | 1.00 (0.86, 1.16) | 1.02 (0.96, 1.09) |

scores were related to increased peer relationship problems. Finally, in girls, higher feminine play behavior scores were associated with decreased likelihood of clinically relevant autistic traits and behavioral difficulties. All abovementioned associations display small effect sizes considering the scales of dependent variables, which is expected within a non-clinical study sample.

The co-occurrence of GD, and to some extent GNC as an indicator of GD, with ASD and autistic traits has been investigated broadly in the last decade [21, 24]. Our results show an association between increased composite play behavior score, that indicates a single domain of GNC in girls, and increased autistic traits, are partly in line with the existing literature. Namely, similar links were demonstrated in a non-clinical study population consisting of children between 6 and 12 years of age [21]. Gender non-conforming expressions in adults with ASD were also reported [17]. In those particular studies, the associations have only been shown for birth-assigned females. Similar to those studies, we did not observe any significance regarding autistic traits in boys. On the other hand, Hull et al. showed that boys with ASD engaged in less gendered play compared to boys without ASD, while there was no significant difference in girls' play based on ASD [16]. Dissimilarities in play behavior between children with and without ASD for both girls and boys were observed [40], with non-significant results found for both genders as well [41]. Those findings diverge from our results; however, one should note that we did not include participants with ASD in oppose to those studies. Having a population-based study sample utilizing symptoms might have led to non-significant results in boys and partly in girls and small effect sizes in general.

We present different patterns of associations between gender-specific play and autistic traits among girls and boys, which should be interpreted in connection to child development. Gender stereotypes become rigid between ages three and five years [42], which extends to gender-specific play [36], and diminish later on. Gender-specific play changes during childhood and the timing of assessment has an impact on these changes, as demonstrated by Hull et al. [16]. In our study, play behavior was evaluated at the age of seven, which may have contributed to capturing reduced gender-specific play and, consequently, non-significant results. Additionally, boys are suggested to exhibit less gender variation in behavior compared with girls due to facing more social pressure to adhere to gender norms [43]. This tendency might also apply to gender-specific play behavior in this current study, potentially contributing to non-significant results among boys. The lack of significant results among birth-assigned males also align with the extreme male brain theory (EMB) [44, 45]. The hypothesis states that higher prenatal testosterone exposure leads to development of a male-typical brain, and this might explain the high co-occurrence of GD and ASD among birth-assigned females. However, evidence for the EMB is inconclusive; opposing findings have been reported [46, 47], and the theory has been critically scrutinized [48]. The EMB seems insufficient to explain the biological underpinnings of the connection between GD and ASD. Even though our results do not conflict with the EMB, they should be interpreted cautiously considering the concerns about the theory and the lack of biological measures in our study.

In recent years, the binary framework of sex and gender has also been started to be disregarded in neurobiology. For instance, the mosaic brain theory has been proposed, which claims that biological sex affects the brain of both females and males through several mechanisms and leads to "femaleness" or "maleness" in different features of the brain [49]. A recent study by Rodin et al. identified specific mosaics in the brains of people with ASD, which implies a less binary interpretation of sex and gender in autism [50]. These findings align with the study by Hull et al., which showed less gendered developmental trajectories of play behavior among children with ASD in comparison to the ones without ASD [16]. Even though our results on autistic traits might also signal less gender-specific play for girls, we did not observe

such a finding for boys. Using scales as such PSAI does not allow for exploring play behavior beyond the binary perspective, and future research should also focus on gender-neutral play respecting the accumulating evidence on play behavior in ASD.

Hypotheses attempting to delineate the link between GNC/GD and autistic traits/ASD are not limited to biological ones, and social and psychological factors have also been proposed [26]. Individuals with ASD exhibit difficulties to conform to social norms and expectations and this aspect of ASD has been proposed to be linked to the development of GNC and GD [51]. Different gender identities have also been stated to be parts of varied perceptions of one's whole identity among persons with ASD [52]. A few researchers have also hypothesized GNC and symptoms of GD as preoccupations and obsessive thoughts in relation to ASD [53, 54], but these hypotheses seem to be limited to case reports. Various biological, social and psychological factors might contribute to the high co-occurrence of GD and ASD, and this might extend to our results on associated symptoms among girls. While potential explanations have mainly been proposed for clinical conditions, it is important to note that our findings on symptoms cannot be generalized to clinical diagnoses. Nonetheless, some of our results support existing evidence and contribute to broadening our understanding, potentially offering better support to children with subthreshold conditions.

Even though the co-occurrence of GD and ASD has been established by several studies as of now [24], potential risk of misclassification of behavioral adversities arising from social difficulties, such as autistic traits, have been claimed, since different measures of gender and behavioral outcomes have been used instead of the actual diagnoses in most of the studies [27]. These are indeed noteworthy statements, and to overcome the potential misclassification of behavioral outcomes, the dependent variables consisted of two different scales in our study (SRS and SDQ), as it has been recommended in the literature [55]. While autistic traits might potentially lead to some behavioral difficulties, examining two sets of symptom clusters helps distinguish the relationships with GNC.

Regarding behavioral difficulties, our findings are consistent with previous studies in which GNC was linked to internalizing or externalizing behavioral difficulties in girls [29, 30]. Moreover, our results display that higher feminine play behavior in boys is related to poorer peer relationships, which also aligns with previous studies [29, 30]. More specifically, gender non-normative patterns of play behavior have been shown to be related to experiencing difficulties in peer relationships among boys and girls [56]. The current literature suggests that the high co-occurrence between GNC and decreased psychological well-being is largely mediated by social intolerance, discrimination and stigmatization toward children with GNC [3, 57]. However, the cross-sectional design of the current study does not allow us to make causal interpretations in the abovementioned associations.

Our results also show associations between masculine and composite scores in play behavior and behavioral difficulties in boys, particularly for conduct problems and hyperactivity. As higher masculine or composite scores are not interpreted as GNC in boys, these results cannot be explained in the context of experiencing gender roles related discrimination and stigma. A reasonable explanation of these results might be that children with higher behavioral difficulties or hyperactivity traits might tend to engage in more masculine play such as rough and tumble play. Similarly, a potential overlap between the constructs of masculine play behavior and particular behavioral difficulties such as hyperactivity should be taken into consideration for the interpretation of the significant results in girls.

The present study has several limitations. First, the cross-sectional design and the lack of time aspects prevent us from drawing conclusions about the directionality between independent variables and dependent variables. Second, the PSAI is a fairly old instrument which was constructed 30 years ago, and it might not completely display children's play behavior

contemporarily; however, this is unlikely to change the association in any direction. Third, we only focused on a single domain of GNC and used play behavior as a proxy, but GNC can certainly be expressed in various ways. Yet, play behavior is emphasized in different diagnostic criteria for GD or gender incongruence in children, which puts its relevance forward within this context. Fourth, all independent and dependent variables were collected through parent-reported scales. Parents might have different tendencies to over- or underestimate their children's behavior, which might have partly resulted in non-differential misclassification of independent and dependent variables. This, however, is unlikely to bias the results in a certain direction. There are also a number of strengths of this study. For instance, the sample size is large in comparison to similar studies; we used a non-clinical study population, which is rare in the literature, and we used different behavioral outcomes, which allowed us to assess the potential relationships more thoroughly. Furthermore, we were able to adjust for parental attitudes toward children's play behavior which controls for the effect parents have on their children's choice of play.

In conclusion, we show that GNC measured via gender-specific play behavior is associated with autistic traits and behavioral difficulties in girls, and with peer relationship problems in boys. Our results suggest that GNC is linked to autistic traits in girls, and different behavioral difficulties in both genders, beyond clinical populations. This highlights the importance of supporting children with GNC starting from early childhood to have more favorable mental health outcomes.

## Supporting information

**S1 Table. Differences between the characteristics of included and excluded study participants.**
(DOCX)

**S2 Table. Unadjusted associations between gender-specific play behavior scores and autistic traits (N = 718).**
(DOCX)

**S3 Table. Unadjusted associations between gender-specific play behavior scores and behavioral outcomes (N = 718).**
(DOCX)

**S4 Table. Unadjusted associations between gender-specific play behavior scores and autistic traits and clinically relevant behavioral outcomes (N = 718).**
(DOCX)

**S1 Fig. Hypothesized confounders in the associations between play behavior and behavioral outcomes.** The green node indicates the independent variable, the blue node the dependent variable. The red nodes indicate potential confounding factors. I indicates the dependent variable.
(TIF)

**S2 Fig. Collinearity scatterplot for SDQ total score and feminine score among boys.**
(TIF)

**S3 Fig. Collinearity scatterplot for SDQ total score and feminine score among girls.**
(TIF)

**S4 Fig. Collinearity scatterplot for SRS total t-score and feminine score among boys.**
(TIF)

**S5 Fig. Collinearity scatterplot for SRS total t-score and feminine score among girls.**
(TIF)

**S1 Questionnaire. PSAI used in the SELMA study.**
(DOCX)

**S2 Questionnaire. The scale on parental attitudes toward play behavior used in the SELMA study.**
(DOCX)

## Acknowledgments

The authors thank the participating families and staff of the SELMA study.

## Author Contributions

**Conceptualization:** Joëlle Rüegg, Carl-Gustaf Bornehag.

**Formal analysis:** Fatih Özel, Marlene Stratmann.

**Funding acquisition:** Joëlle Rüegg, Carl-Gustaf Bornehag.

**Methodology:** Fotios C. Papadopoulos, Carl-Gustaf Bornehag.

**Project administration:** Carl-Gustaf Bornehag.

**Supervision:** Joëlle Rüegg, Carl-Gustaf Bornehag.

**Writing – original draft:** Fatih Özel, Marlene Stratmann.

**Writing – review & editing:** Fatih Özel, Marlene Stratmann, Fotios C. Papadopoulos, Joëlle Rüegg, Carl-Gustaf Bornehag.

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
