## [Decision Letter · Decision Letter 0]

2 May 2024

PONE-D-23-43003Gender-specific play in relation to autistic traits and behavioral difficulties at the age of seven in the SELMA studyPLOS ONE

Dear Dr. Stratmann,

Thank you for submitting your manuscript to PLOS ONE. After careful consideration, we feel that it has merit but does not fully meet PLOS ONE’s publication criteria as it currently stands. Therefore, we invite you to submit a revised version of the manuscript that addresses the points raised during the review process.

We look forward to receiving your revised manuscript.

Kind regards,

Ewa Pisula

Academic Editor

PLOS ONE

Journal Requirements:

2. Thank you for stating the following financial disclosure: "This work received funding from the European Union’s Horizon 2020 research and innovation

program under grant agreement no. 634880, EDC-MixRisk"

Additional Editor Comments :

The article was thoroughly reviewed by two reviewers who made essential comments. They both appreciated the merit, although their opinions on the value of this work varied. Both reviewers pointed out several weaknesses in the paper, precisely suggesting specific changes. I fully agree with Reviewer 2 that the discussion section seems superficial and poorly theoretically grounded. The suggestion to refer to a paper by Hull et al. (2022) is also very valuable. Because the study has some strengths, its results are worth presenting, and the paper can be significantly improved according to the reviewers' comments, my decision is a major revision.

Reviewers' comments:

Reviewer's Responses to Questions

**Comments to the Author**

1. Is the manuscript technically sound, and do the data support the conclusions?

Reviewer #1: Partly

Reviewer #2: Yes

2. Has the statistical analysis been performed appropriately and rigorously? 

Reviewer #1: Yes

Reviewer #2: Yes

3. Have the authors made all data underlying the findings in their manuscript fully available?

Reviewer #1: No

Reviewer #2: No

4. Is the manuscript presented in an intelligible fashion and written in standard English?

Reviewer #1: Yes

Reviewer #2: Yes

5. Review Comments to the Author

Reviewer #1: Review of the article Gender-specific Play in Relation to autistic traits and Behavioral Difficulties at the Age of Seven in the SELMA study

Thank you for the opportunity to review this manuscript.

The article reports the results of a study examining the relationship between feminine vs. masculine types of play in children and their autistic traits and problem behaviors. The feminine/masculine play scores were used as a proxy for gender nonconformity. Assessments were based on parental reports. The design of the study was correlational. The Authors found gender-specific differences in the effects they observed. Overall, I think this work may make a valuable contribution, but it requires improvement. I list my concerns below.

Introduction

Line 76 Please define the term “gender diversity.”

Line 104 There seems to be a problem with this sentence. Did you mean “that shed light” instead of “to shed light”?

Results

Description of the results and Table 1 Please provide the effect sizes where applicable. Regarding the result indicating higher SRS scores in girls than boys, please give a possible explanation in the Discussion section. Please clarify whether subscale scores in SES and SDQ subscales differed significantly in boys and girls.

Results of regression analyses Please report collinearity statistics. Please provide scatterplots to demonstrate a linear association between variables, which is the key assumption for conducting linear regression. Please discuss the effect sizes.

Tables 2 and 3 are cut.

Discussion

Line 294 Please provide references.

Lines 302-304 I think that drawing such firm conclusions based on null results (here: non-significant associations) is an overstatement.

Lines 314-325 How are these considerations related to your study?

Lines 326-341 Here, I am confused, as the study involved the general population and did not really focus on a sample of children with a diagnosis of ASD.

Lines 363-365 Clearly, the directionality cannot be determined in a correlational study like this one. Would it not be better to use the term “dependent variable” instead of “outcome” throughout the article?

Please discuss the use of parental evaluation of play type as a proxy for gender nonconformity. Is this a sufficient indicator?

Overall, the results of the study have not been adequately and thoroughly discussed. For instance, “social and psychological factors” (line 327) have been vaguely mentioned but not addressed regarding possible mechanisms of gender differences in the present study.

Supplementary Figure 1 Please add an annotation explaining what “I” means.

Reviewer #2: My thanks for the opportunity to read this paper exploring gender-specific play and the relationship with autistic traits and problem behaviours. The authors found gender-atypical play was associated with higher autistic traits in girls, and with subtypes of behavioural difficulties in both boys and girls. Although this paper does have strengths, such as exploring these associations in a large, population-based sample, there are several limitations which for me detract from its impact. Notably, the paper is missing engagement with a recent study exploring a very similar research question, which raised interesting hypotheses around the development of gender-specific play, and which could extent the discussion of the present paper. Additionally, the discussion section felt quite descriptive to me and did not really address any theories as to why the observed differences/lack thereof may have been observed. In particular I would have liked to see more consideration of how autistic traits and problem behaviours might be related to each other or distinct, and how gender-specific play interacts with both of these in childhood. The paper requires some reworking to make a clear contribution to current understanding.

Comments:

Abstract – ‘higher composite scores’ is quite hard to interpret for someone who doesn’t know the measure – could this be described more precisely (e.g. “increased masculine/decreased feminine play”)?

What exactly do the conclusions mean for clinical settings or for our understanding of gender development? It’s not clear what the impact of these findings would be from the abstract.

Funding – please supply more information regarding who the funding was awarded to and whether the funders played a role in any stage of the study.

Introduction

It is good that the authors distinguish between gender non-confirming and gender dysphoria. However I’d like to see some more consideration of how gender-specific play fits within these categories – how do these authors define this, and why is gender-specific play an appropriate proxy for gender non-confirming behaviour in this sample?

There is a relatively recent study (Hull, Heuvelman et al., 2022; DOI: 10.1177/13623613221139373) that explored precisely this question of gender-specific play and the association with autism within longitudinal population-level data. I think it is important that the authors engage with this study, particularly the age-related findings for boys, in both the introduction and discussion of this article.

Methods

Some more detail about the SDQ would be helpful, such as describing the different subscales and how they are interpreted – it might be helpful, for instance, to note that the prosocial behaviour subscale is scored in reverse to the others, with higher scores representing fewer behavioural difficulties.

Were the same confounders proposed for both autistic trait and problem behaviour outcomes? What was the rationale behind this?

Results

Is there any information on how representative the participants were of the entire study population? E.g. did included versus excluded/missing data participants differ on any key variables or characteristics?

Tables 2 and 3 were cut off for me and so I could not read all results. Please could these be reformatted to landscape orientation or presented on separate pages.

Please also present the full model fit statistics for each model somewhere (e.g. in supplementary material).

Discussion

What do these findings actually mean? So far the discussion seems very descriptive with little consideration of how this relates to either theory around child development, or supporting children with autistic traits or behavioural difficulties. In particular, I’d like to see more discussion of why different patterns of associations were found in boys and girls – which factors might be driving this?

Again I think it would be helpful to refer to the paper by Hull et al. (2022) which engages with the Extreme Male Brain theory and other theories of gender development in autism, and advocates for a developmental approach to these theories. In addition, this paper found that the age at which play behaviours are measured might impact gender typicality for boys; this could be discussed here.

The authors refer very briefly to the idea of moving beyond a binary concept of sex and gender, but could expand upon this. I wonder again how much the binary measure used (PSAI) plays a role in this – it is not possible to look at neutral/agender behaviours if all play behaviours are categorised as only male or female. How would the authors recommend research explore beyond binary sex/gender in the future?

I didn’t quite follow the argument made in the first paragraph of page 17. Are the authors suggesting that the SDQ and SRS are measuring the same construct? If so, was this autistic traits or problem behaviours? And why were all the results presented separately?

6. PLOS authors have the option to publish the peer review history of their article (what does this mean?). If published, this will include your full peer review and any attached files.

Reviewer #1: **Yes: **Alicja Niedźwiecka

Reviewer #2: No

---

## [Author Response · Author response to Decision Letter 0]

20 Jun 2024

Dear Dr. Pisula, Dr. Niedźwiecka and reviewers,

We appreciate the opportunity to review our manuscript. We believe that the manuscript improved with your ideas and suggestions and hope that we addressed them to your satisfaction. Please see our point-by-point responses below.

Best regards,

Marlene Stratmann

Journal Requirements:

Response: Thank you for reminding us of the requirements. We have updated the manuscript and supporting information according to the guidelines.

2. Thank you for stating the following financial disclosure: "This work received funding from the European Union’s Horizon 2020 research and innovation program under grant agreement no. 634880, EDC-MixRisk". Please state what role the funders took in the study. If the funders had no role, please state: "The funders had no role in study design, data collection and analysis, decision to publish, or preparation of the manuscript." If this statement is not correct you must amend it as needed. Please include this amended Role of Funder statement in your cover letter; we will change the online submission form on your behalf.

Response: We have updated the cover letter accordingly.

Response: We have added contact information to the data availability statement in the submission system in case someone wants to request the data. We also explained that there are legal and ethical restrictions according to the General Data Protection Regulation, the Swedish law, the Swedish Data Protection Act, and the Swedish Ethical Review Act, as data can be used to potentially identify study participants and the data contains sensitive information.

Response: Restrictions apply to data availability; please refer to the previous comment.

Response: Thank you for bringing our attention to this. We updated the in-text citations and the captions according to the guidelines.

 Additional Editor Comments:

5. The article was thoroughly reviewed by two reviewers who made essential comments. They both appreciated the merit, although their opinions on the value of this work varied. Both reviewers pointed out several weaknesses in the paper, precisely suggesting specific changes. I fully agree with Reviewer 2 that the discussion section seems superficial and poorly theoretically grounded. The suggestion to refer to a paper by Hull et al. (2022) is also very valuable. Because the study has some strengths, its results are worth presenting, and the paper can be significantly improved according to the reviewers' comments, my decision is a major revision.

Response: Thank you so much for your evaluation. All the comments have proven helpful and we changed the manuscript to the best of our ability to hopefully make the discussion more theoretically founded and deeper. We have also included the study by Hull et al. (2022), as it is perfectly connected to our research.

Reviewer # 1

Review of the article Gender-specific Play in Relation to autistic traits and Behavioral Difficulties at the Age of Seven in the SELMA study

Thank you for the opportunity to review this manuscript.

The article reports the results of a study examining the relationship between feminine vs. masculine types of play in children and their autistic traits and problem behaviors. The feminine/masculine play scores were used as a proxy for gender nonconformity. Assessments were based on parental reports. The design of the study was correlational. The Authors found gender-specific differences in the effects they observed. Overall, I think this work may make a valuable contribution, but it requires improvement. I list my concerns below.

 Comments

Introduction

1. Line 76 Please define the term “gender diversity.”

Response: We included the definition of gender diversity and it now is: “Still, the proportion of transgender and gender diverse persons, i.e., people whose gender identities and/or expressions are different from sociocultural norms, in the general population is estimated to be between 0.02 and 8.4% [7].”

2. Line 104 There seems to be a problem with this sentence. Did you mean “that shed light” instead of “to shed light”?

Response: Thank you for pointing out this error. We have updated the sentence according to your suggestions.

 Results

3. Description of the results and Table 1 Please provide the effect sizes where applicable.

Response: Thank you for this comment. As we only present the descriptive characteristics of our study population and comparisons of those characteristics between girls and boys are not our main research questions, we decided not to show effect sizes or p-values in Table 1. However, we present effect sizes and p-values for PSAI, SRS, and SDQ in Supplementary Table 1 considering those comparisons are relevant to the main research questions and may help readers to interpret the main results better. We added a sentence to the method section to point to the supplementary material.

4. Regarding the result indicating higher SRS scores in girls than boys, please give a possible explanation in the Discussion section. Please clarify whether subscale scores in SES and SDQ subscales differed significantly in boys and girls.

Response: Thank you for your comment. As explained in the previous comments, p-values are now presented in Supplementary Table 1. The result indicating higher SRS scores among girls compared with boys is not significant. Since we used standardized SRS scores, which are adjusted for sex differences, this marginal non-significant difference should not be a concern in a population-based study sample. For this reason, we did not elaborate on that finding in the discussion section. 

5. Results of regression analyses Please report collinearity statistics. Please provide scatterplots to demonstrate a linear association between variables, which is the key assumption for conducting linear regression. Please discuss the effect sizes.

Response: As we examined several associations with linear regression analysis, we exemplified collinearity scatterplots for some of the associations in Supplementary Figures 2-5. Collinearity statistics are provided in the extra file for regression analysis along with the full model fit statistics.

Based on your suggestions, we put more emphasis on the effects sizes in the discussion section as follows: “All abovementioned associations display small effect sizes considering the scales of dependent variables, which is expected within a non-clinical study sample.” And further: “Having a population-based study sample utilizing symptoms might have led to non-significant results in boys and partly in girls and small effect sizes in general.” 

6. Tables 2 and 3 are cut.

Response: We apologise for this. According to the journal guidelines regarding size of tables we referred to this rule: “Do not split your table or otherwise try to make the table appear within the manuscript margins if it does not fit on one page. In Word, tables that run off of the manuscript page can be seen using Draft View. In the PDF version of the published article, very wide tables may be printed sideways, and long tables may span more than one page.” We have changed this to make the tables readable in non-draft view in Word.

 Discussion

7. Line 294 Please provide references.

Response: We included references following your suggestion. 

8. Lines 302-304 I think that drawing such firm conclusions based on null results (here: non-significant associations) is an overstatement.

Response: We toned our statement down as follows: “Similar to those studies, we did not observe any significance regarding autistic traits in boys.” 

9. Lines 314-325 How are these considerations related to your study?

Response: Thank you for your comment. We adjusted that section and emphasized its relation to our study as follows:

“In recent years, the binary framework of sex and gender has also been started to be disregarded in neurobiology. For instance, the mosaic brain theory has been proposed, which claims that biological sex affects the brain of both females and males through several mechanisms and leads to “femaleness” or “maleness” in different features of the brain [49]. A recent study by Rodin et al. identified specific mosaics in the brains of people with ASD, which implies a less binary interpretation of sex and gender in autism [50]. These findings align with the study by Hull et al., which showed less gendered developmental trajectories of play behavior among children with ASD in comparison to the ones without ASD [16]. Even though our results on autistic traits might also signal less gender-specific play for girls, we did not observe such a finding for boys. Using scales as PSAI does not allow for exploring play behavior beyond the binary perspective, and future research should also focus on gender-neutral play respecting the accumulating evidence on play behavior in ASD.” 

10. Lines 326-341 Here, I am confused, as the study involved the general population and did not really focus on a sample of children with a diagnosis of ASD.

Response: In that paragraph, we aim to present potential explanation not only for GD and ASD but also for GNC and autistic tratis. We adjusted the text not to lead any confusion and put more emphasis on the fact that we only investigated symptom dimensions: “Hypotheses attempting to delineate the link between GNC/GD and autistic traits/ASD are not limited to biological ones, and social and psychological factors have also been proposed [26].” And further: “While potential explanations have mainly been proposed for clinical conditions, it is important to note that our findings on symptoms cannot be generalized to clinical diagnoses…” 

11. Lines 363-365 Clearly, the directionality cannot be determined in a correlational study like this one. Would it not be better to use the term “dependent variable” instead of “outcome” throughout the article?

Response: Thank you for your suggestion. We replaced the term “outcome variable” with “outcome variable” throughout the text.

12. Please discuss the use of parental evaluation of play type as a proxy for gender nonconformity. Is this a sufficient indicator?

Response: We improved our statement in the discussion section concerning the potential limitations of using a parent-reported play behavior as a proxy for gender nonconformity: “Third, we only focused on a single domain of GNC and used play behavior as a proxy, but GNC can certainly be expressed in various ways. Yet, play behavior is emphasized in different diagnostic criteria for GD or gender incongruence in children, which puts its relevance forward within this context. Fourth, all independent and dependent variables were collected through parent-reported scales. Parents might have different tendencies to over- or underestimate their children's behavior, which might have partly resulted in non-differential misclassification of independent and dependent variables.” 

13. Overall, the results of the study have not been adequately and thoroughly discussed. For instance, “social and psychological factors” (line 327) have been vaguely mentioned but not addressed regarding possible mechanisms of gender differences in the present study.

Response: Thank you for your suggestion. We aimed to improve the discussion section following your comments. Specifically, we elaborated on social and psychological factors as follows: “Hypotheses attempting to delineate the link between GNC/GD and autistic traits/ASD are not limited to biological ones, and social and psychological factors have also been proposed [26]. Individuals with ASD exhibit difficulties to conform to social norms and expectations and this aspect of ASD has been proposed to be linked to the development of GNC and GD [51]. Different gender identities have also been stated to be parts of varied perceptions of one’s whole identity among persons with ASD [52]. A few researchers have also hypothesized GNC and symptoms of GD as preoccupations and obsessive thoughts in relation to ASD [53, 54], but these hypotheses seem to be limited with case reports. Various biological, social and psychological factors might contribute to the high co-occurrence of GD and ASD, and this might extend to our results on associated symptoms among girls…” 

14. Supplementary Figure 1 Please add an annotation explaining what “I” means.

Response: We added an explanation in the figure, which is: “The green node indicates the independent variable, the blue node the dependent variable. The red nodes indicate potential confounding factors.” The “I” in the DAG does not mean anything specific but is a choice from the developers of dagitty.net to indicate the dependent variable. We hope that the added explanation is sufficient.

Reviewer #2

My thanks for the opportunity to read this paper exploring gender-specific play and the relationship with autistic traits and problem behaviours. The authors found gender-atypical play was associated with higher autistic traits in girls, and with subtypes of behavioural difficulties in both boys and girls. Although this paper does have strengths, such as exploring these associations in a large, population-based sample, there are several limitations which for me detract from its impact. Notably, the paper is missing engagement with a recent study exploring a very similar research question, which raised interesting hypotheses around the development of gender-specific play, and which could extent the discussion of the present paper. Additionally, the discussion section felt quite descriptive to me and did not really address any theories as to why the observed differences/lack thereof may have been observed. I

---

## [Decision Letter · Decision Letter 1]

29 Jul 2024

Gender-specific play behavior in relation to autistic traits and behavioral difficulties at the age of seven in the SELMA study

PONE-D-23-43003R1

Dear Dr. Stratmann,

We’re pleased to inform you that your manuscript has been judged scientifically suitable for publication and will be formally accepted for publication once it meets all outstanding technical requirements.

Kind regards,

Ewa Pisula

Academic Editor

PLOS ONE

Additional Editor Comments (optional):

Dear Authors, congratulations on the final version of the text and its acceptance for publication in PLOS ONE. As you can see, Reviewer 2 made a comment about the use of the term "“children with gender nonconformity", suggesting the use of the more inclusive term "gender non-conforming children". I agree with the Reviewer, but I leave it up to you to decide on this matter.

Reviewers' comments:

Reviewer's Responses to Questions

**Comments to the Author**

1. If the authors have adequately addressed your comments raised in a previous round of review and you feel that this manuscript is now acceptable for publication, you may indicate that here to bypass the “Comments to the Author” section, enter your conflict of interest statement in the “Confidential to Editor” section, and submit your "Accept" recommendation.

Reviewer #1: All comments have been addressed

Reviewer #2: All comments have been addressed

2. Is the manuscript technically sound, and do the data support the conclusions?

Reviewer #1: Yes

Reviewer #2: Yes

3. Has the statistical analysis been performed appropriately and rigorously? 

Reviewer #1: Yes

Reviewer #2: Yes

4. Have the authors made all data underlying the findings in their manuscript fully available?

Reviewer #1: No

Reviewer #2: No

5. Is the manuscript presented in an intelligible fashion and written in standard English?

Reviewer #1: Yes

Reviewer #2: Yes

6. Review Comments to the Author

Reviewer #1: I appreciate the Authors addressing my comments so thoroughly.

I have one minor comment:

Page 3, lines 58-59: Instead of using the phrase “children with gender nonconformity”, I suggest using more inclusive language, e.g. “gender non-conforming children.”

https://www.hrc.org/resources/supporting-your-young-gender-non-conforming-child

Reviewer #2: My thanks to the authors for addressing my comments. The manuscript is clearer and better justified as a result, particularly the discussion where the authors thoughtfully engage with the Extreme Male Brain theory and Mosaic Brain theory. I have no further comments.

7. PLOS authors have the option to publish the peer review history of their article (what does this mean?). If published, this will include your full peer review and any attached files.

Reviewer #1: **Yes: **Alicja Niedźwiecka

Reviewer #2: No

---

## [Editor Report · Acceptance letter]

1 Aug 2024

PONE-D-23-43003R1 

PLOS ONE

Dear Dr. Stratmann, 

I'm pleased to inform you that your manuscript has been deemed suitable for publication in PLOS ONE. Congratulations! Your manuscript is now being handed over to our production team.

Kind regards, 

on behalf of

Dr. Ewa Pisula 

Academic Editor

PLOS ONE